# A New Pentafluorothio-Substituted Curcuminoid with Superior Antitumor Activity

**DOI:** 10.3390/biom11070947

**Published:** 2021-06-25

**Authors:** Benedikt Linder, Leonhard H. F. Köhler, Lisa Reisbeck, Dominic Menger, Dharmalingam Subramaniam, Christel Herold-Mende, Shrikant Anant, Rainer Schobert, Bernhard Biersack, Donat Kögel

**Affiliations:** 1Experimental Neurosurgery, Frankfurt University Hospital, Theodor-Stern-Kai 7, 60590 Frankfurt am Main, Germany; Lisa.Henkel@kgu.de (L.R.); dominic.menger@gmx.de (D.M.); koegel@em.uni-frankfurt.de (D.K.); 2Organic Chemistry Laboratory, University of Bayreuth, Universitätsstrasse 30, 95440 Bayreuth, Germany; Leonhard.Koehler@uni-bayreuth.de (L.H.F.K.); rainer.schobert@uni-bayreuth.de (R.S.); 3Cancer Biology Department, University of Kansas Medical Center, 3901 Rainbow Boulevard, Kansas City, MO 66160, USA; dsubramaniam@kumc.edu (D.S.); sanant@kumc.edu (S.A.); 4Department of Neurosurgery, Division of Experimental Neurosurgery, University Hospital Heidelberg, INF 400, 69120 Heidelberg, Germany; Christel.Herold-Mende@med.uni-heidelberg.de; 5German Cancer Consortium (DKTK), Partner Site Frankfurt, 60590 Frankfurt am Main, Germany

**Keywords:** curcumin, piperidone, pentafluorothio group, fluorine, anticancer agents, cell death, glioblastoma

## Abstract

A new and readily available pentafluorothiophenyl-substituted N-methyl-piperidone curcuminoid **1a** was prepared and investigated for its anti-proliferative, pro-apoptotic and cancer stem cell-differentiating activities against a panel of human tumor cell lines derived from various tumor entities. The compound **1a** was highly anti-proliferative and reached IC_50_ values in the nanomolar concentration range. **1a** was superior to the known anti-tumorally active curcuminoid EF24 (**2**) and its known N-ethyl-piperidone analog **1b** in all tested tumor cell lines. Furthermore, **1a** induced a noticeable increase of intracellular reactive oxygen species in HT-29 colon adenocarcinoma cells, which possibly leads to a distinct increase in sub-G1 cells, as assessed by cell cycle analysis. A considerable activation of the executioner-caspases 3 and 7 as well as nuclei fragmentation, cell rounding, and membrane protrusions suggest the triggering of an apoptotic mechanism. Yet another effect was the re-organization of the actin cytoskeleton shown by the formation of stress fibers and actin aggregation. **1a** also caused cell death in the adherently cultured glioblastoma cell lines U251 and Mz54. We furthermore observed that **1a** strongly suppressed the stem cell properties of glioma stem-like cell lines including one primary line, highlighting the potential therapeutic relevance of this new compound.

## 1. Introduction

Curcumin (compound **3**) is a simple phenolic natural product isolated from the rhizome of the plant *Curcuma longa*. Though displaying promising anticancer activities, its potential as a drug candidate is limited by its rapid metabolic degradation, its low water solubility and its insufficient bioavailability both in plasma and in tissues. [1,2] Thus, new curcumin derivatives with improved anticancer and pharmacokinetic properties are sought for. Curcuminoids based on the bis(benzylidene)piperidone scaffold such as EF24 (compound **2**) have shown improved anticancer activities when compared to curcumin as well as superior bioavailability, stability and uptake rates (Figure 1). [3,4,5,6,7,8,9] More recently, various halogenated bis(benzylidene)piperidone derivatives with pronounced anticancer activities as well as distinct anti-angiogenic activities were reported. [10,11] In particular, the pentafluorothio group (-SF_5_) has attracted certain interest concerning drug discovery and has shown distinct potential for the design of new anticancer active curcuminoids such as **1b** in an initial work of our group (Figure 1) [11,12]. The SF_5_ group is a chemically stable xenobiotic mimic of negatively charged biomolecules and it was labelled as “super trifluoromethyl group”, which is often superior to more common fluoro and trifluoromethyl substituents in bioactive compounds [12]. Our working hypothesis is, that **1a** exceeds in its anticancer effects those of **1b** and **2**. Our primary aim is to test this hypothesis across several tumor cell lines derived from multiple tumor entities. Herein, we report on the anticancer properties of a new and superior pentafluorothio-substituted bis(benzylidene)piperidone derivative **1a** with high potential as a future anticancer drug (Figure 1).

## 2. Materials and Methods

### 2.1. Chemistry

#### 2.1.1. General

Starting materials and reagents were purchased from Sigma-Aldrich. The known compounds EF24 (**2**) and **1b** were synthesized according to literature procedures. [10,13] The following instruments were used: melting points (uncorrected), Gallenkamp; IR spectra, Perkin-Elmer Spectrum One FT-IR spectrophotometer with ATR sampling unit; nuclear magnetic resonance spectra, BRUKER Avance 300 spectrometer; chemical shifts are given in parts per million (δ) downfield from tetramethylsilane as internal standard; mass spectra, Varian MAT 311A (EI), Thermo Fisher Scientific Q Exactive (ESI-HRMS).

#### 2.1.2. (E)-1-Methyl-3,5-bis(4-pentafluorothiobenzylidene)-4-piperidone (**1a**)

1-Methyl-4-piperidone (85 mg, 0.75 mmol) was dissolved in MeOH (5 mL) and 4-(pentafluorothio)-benzaldehyde (348 mg, 1.5 mmol) was added. NaOH (40 mg, 1.0 mmol) and H_2_O (1 mL) were added, and the reaction mixture was stirred at room temperature for 1 h. The formed precipitate was collected, washed with MeOH and dried in vacuum. Yield: 230 mg (0.43 mmol, 57%); yellow solid of m.p. 218–220 °C; *ν*_max_(ATR)/cm^−1^ 2985, 2954, 2856, 2781, 1678, 1625, 1593, 1574, 1493, 1459, 1410, 1387, 1333, 1294, 1275, 1180, 1127, 1100, 1059, 993, 926, 837, 823, 811, 799, 728, 703, 661; ^1^H NMR (300 MHz, DMSO-d_6_) δ2.38 (3 H, s), 3.75 (4 H, s), 7.65 (2 H, s), 7.71 (4 H, d, J = 8.6 Hz), 7.98 (4 H, d, J = 8.6 Hz); ^13^C NMR (75.5 MHz, DMSO-d_6_) δ45.1, 56.0, 126.1, 131.1, 132.6, 136.0, 138.5, 152.5, 186.8; *m/z* (EI) 541 (100) [M^+^], 540 (42), 513 (41), 512 (41), 414 (11), 310 (22), 296 (23), 270 (45), 242 (8), 143 (7), 115 (13), 43 (14); HRMS (ESI) C_20_H_18_F_10_NOS_2_ [M + H]^+^ calcd. 542.06709, found 542.06612. 

### 2.2. Biological Evaluation

#### 2.2.1. Cell Culture Conditions

518A2 (Department of Radiotherapy, Medical University of Vienna, Vienna, Austria) [14] melanoma, Panc-1 (ACC-783) pancreatic ductular adenocarcinoma, KB-V1^Vbl^ (ACC-149) cervix carcinoma, MCF-7^Topo^ (ACC-115) breast carcinoma, HT-29 (ACC-299), HCT-116 (ACC-581), DLD-1 (ACC-278), SW-480 (CCL-228) colon carcinoma, were cultivated in Dulbecco’s Modified Eagle Medium (DMEM) supplemented with 10% fetal bovine serum, and 1% antibiotic-antimycotic at 37 °C, 5% CO_2_, and 95% humidity. To keep MCF-7^Topo^ and KB-V1^Vbl^ cells resistant, the maximum-tolerated dose of topotecan or vinblastine were respectively added to the cell culture medium 24 h after every passage. The cells were used in passages between 40 and 47 (518A2), 102 and 110 (Panc-1), 223 and 229 (KB-V1^Vbl^), 41 and 46 (MCF-7), 34 and 51 (HT-29), 67 and 73 (HCT-116), 51 and 56 (DLD-1), as well as 43 and 48 (SW-480). U251 (formerly known as U-373 MG; ECACC 09063001) [15], Mz54 (CVCL_M406) [16] human glioblastoma cell lines were cultured in DMEM+GlutaMax supplemented with 10% fetal bovine serum and 1% penicillin/streptomycin mixture. The human glioma cell line Mz54 was obtained from the Dept. of Neurosurgery, University Medical Centre, Johannes Gutenberg University Mainz, Germany, where this line was isolated from a recurrent grade IV glioblastoma [16].U251 and Mz54 cells were used in passages between 10 and 25 after re-authentication and 40 to 55 after culture establishment, respectively. The glioma stem-like cells GS-5 [17], NCH644 [18] and the primary culture 17/02 [19] were cultured as floating spheres in Neurobasal A Medium (Gibco, Darmstadt, Germany) containing B27 Supplement (Gibco), 100 U/mL Penicillin 100 µg/mL Streptomycin (Gibco), 1x Glutamax (Gibco), 1x B27 Supplement (Gibco) and 20 ng/mL epidermal growth factor (EGF, Peprotech, Hamburg, Germany) and fibroblast growth factor (FGF, Peprotech). Cells were grown at 5% CO_2_, and 95% humidity. GS-5, NCH644 and 17/02 were used in passages ranging from 8 to 25, 69–82 and 10–22 after culture establishment, respectively. Only mycoplasma-free cell cultures were used. Table 1 shows a summary of all the different cancer entities analyzed.

#### 2.2.2. MTT Assay

The assay was performed as described previously. [20] Briefly, non-GBM cells (5 × 10^4^ cells/mL, 100 µL/well) were grown in 96-well plates for 24 h. Then, they were treated with various concentrations of the test compounds, or vehicle (DMSO, or EtOH) for 72 h at 37 °C. After the addition of 12.5 µL of a 0.5% MTT solution in PBS the cells were incubated for 3 h at 37 °C so that the water-soluble MTT could be converted to formazan crystals. Then, the plates were centrifuged (300× *g*, 5 min, 4 °C), the medium withdrawn, and the formazan dissolved in 25 µL of DMSO containing 10% SDS and 0.6% acetic acid for at least 2 h at 37 °C. Adherently grown GBM cells were seeded at 5 × 10^4^ cells/mL and floating spheres at 8 × 10^4^ cells/mL in 100 µL/well and incubated as above for 48 h or 72 h at 37 °C. Afterwards 10 µL of a 5 mg/mL MTT solution in PBS was added to the cells for 3 h at 37 °C. Then, the spheres were centrifuged shortly to collect the cells on the bottom of the plate and the medium was removed by careful pipetting for both adherently grown and floating sphere GBM cultures. Afterwards, formazan was dissolved in a mixture (24:1 *v*:*v*) of isopropanol and 1 M HCl using 100 µL for adherent cells and 150 µL for GSCs by shaking the plates for at least 20 min. The absorbance of formazan (λ = 570 nm), and background (λ = 630 nm) was measured with a microplate reader (Tecan Spark, Tecan Deutschland GmbH, Crailsheim, Germany). For non-GBM cells the IC_50_ values were derived from dose-inhibition curves as the means ± SD of four independent experiments with respect to vehicle treated control cells set to 100%. For GBM cells the data were first baseline-corrected using blank wells (containing isopropanol:HCl) and then normalized from the DMSO to the maximal inhibition. These data were used for a curve-fitting using the function “log(inhibitor) vs. normalized response—Variable slope” without constraints (GraphPad Prism 7).

#### 2.2.3. Hexoseaminidase Enzyme Assay

To assess proliferation, HCT-116 or SW-480 colon cancer cells were seeded onto 96-well plates (5 × 10^4^ cells/mL, 100 µL/well) and allowed to adhere and grow overnight in 10% heat-inactivated FBS containing DMEM. The cells were then treated with increasing doses of the test compounds in 10% FBS containing DMEM. Analysis of cell proliferation was performed by enzymatic assay as described previously. [21] The IC_50_-values were determined using the curve-fitting function “log(inhibitor) vs. normalized response—Variable slope” (GraphPad Prism).

#### 2.2.4. Zebrafish Angiogenesis Assay

Transgenic Tg(*fli1a:EGFP*) zebrafish with *casper* mutant background were raised under standard conditions at about 28 °C. [22,23] 2 h past fertilization (hpf), the eggs were transferred to petri dishes with E3 medium (5 mM NaCl, 0.17 mM KCl, 0.33 mM CaCl, 0.33 mM MgSO_4_, 0.01% methylene blue, pH 7.2). 24 hpf, the embryos were manually dechorionated, distributed in 6-well plates with 5 mL E3 medium (5 embryos each well) and treated with substances **1a**, **1b**, **2** (10 µM), Axitinib (1 µM), or vehicle (DMSO) for 48 h. The vascular development was determined using the SIV (subintestinal veins) area and documented by fluorescence microscopy (λ_ex_: 488 nm, λ_em_: 509 nm; Leica MZ10F with Zeiss AxioCam Mrc and Mrc-ZEN pro 2012 software). The SIV-area of at least 20 identically treated embryos was quantified with ImageJ software as means ± SD with vehicle control set to 100%. Significant deviations from vehicle treated embryos determined using a *t*-test; *: *p* < 0.05; **: *p* < 0.01; ***: *p* < 0.001; ****: *p* < 0.0001, One-way ANOVA with Dunnett’s multiple comparison test (GraphPad Prism 7).

#### 2.2.5. Caspase-3/7 Activity Assay

Caspase-3/7 activity was measured using the Apo-One^®^ Homogenous Caspase-3/7 Assay Kit (Promega). HT29 colorectal adenocarcinoma cells (2 × 10^5^ cells/mL, 67.5 µL/well) were grown in black 96-well plates for 24 h (37 °C, 5% CO_2_, 95% humidity). The cells were incubated with **1a**, **1b**, **2** (10 µM), positive control Staurosporine (2 µM) or solvent (DMSO) for 6 h under cell culture conditions. After addition of fluorogenic caspase-3/7 substrate solution and 60 min incubation at room temperature, the fluorescence intensity (λ_ex_: 485 nm, λ_em_: 521 nm) was measured using a microplate reader (Tecan F200). To reduce background signals, the blank values without cells were subtracted. Cell vitality was reviewed via MTT-assay as described above (Section 2.2.2). The measurements were performed in triplicate and quoted as means ± SD.

#### 2.2.6. DCFH-DA Assay

HT-29 colorectal adenocarcinoma cells (2 × 10^5^ cells/mL, 3 mL/well) were grown in black 96-well cell culture plates for 24 h (37 °C, 5% CO_2_, 95% humidity). The medium was exchanged with 20 µM DCFH-DA (2′,7′-dichlorohydrofluorescein diacetate) solution in serum-free DMEM and incubated for 30 min (37 °C, 5% CO_2_, 95% humidity). The cells were washed twice with 100 µL PBS and test compounds **1a**, **1b** and **2** in serum free DMEM were added in various concentrations (0.5, 1, 2 µM). After 1h incubation (37 °C, 5% CO_2_, 95% humidity) the cells were washed with PBS again and DCF fluorescence (λ_ex_: 485 nm, λ_em_: 535 nm) was measured with a TECAN microplate reader. Untreated cells (without DCFH-DA solution) acted as blank values and were subtracted from all other measurements. Solvent-treated cells were taken as negative controls and the fluorescence was scaled to 100%. A loss of cell viability was taken into account by performing an MTT-assay as described above (Section 2.2.2). Significant deviations from vehicle treated control was determined using a *t*-test; *: *p* < 0.05; **: *p* < 0.01; ***: *p* < 0.001; ****: *p* < 0.0001, One-way ANOVA with Dunnett’s multiple comparison test (GraphPad Prism 7).

#### 2.2.7. Cell-Cycle Analysis

HT-29 colorectal adenocarcinoma cells (5 × 10^4^ cells/mL, 3 mL/well) were grown in 6-well cell culture plates for 24 h (37 °C, 5% CO_2_, 95% humidity). The treatment with the test compounds **1a**, **1b**, **2** and the vehicle DMSO was carried out for another 24 h (37 °C, 5% CO_2_, 95% humidity). After trypsinization and centrifugation (5 min, 300× *g*, 4 °C), the cells were fixed in 70% EtOH (24 h, 4 °C). The cells were washed with PBS and stained with propidium iodide solution (50 µg/mL PI, 0.1% sodium citrate, 50 µg/mL RNAse A in PBS) for 30 min at 37 °C. DNA-content of 10,000 single cells was determined by fluorescence measurement at λ_em_: 570 nm (λ_em_: 488 nm laser source) with a Beckmann Coulter Cytomics FC500 flow cytometer. The cell cycle phases (sub-G1, G1, S, G2/M) were determined by CXP software (Beckman Coulter, Krefeld, Germany).

#### 2.2.8. Immunofluorescence Staining of Actin Cytoskeleton

HT-29 colorectal adenocarcinoma cells (2 × 10^5^ cells/mL, 0.5 mL/well) were seeded on coverslips in 24-well cell culture plates and incubated for 24 h (37 °C, 5% CO_2_, 95% humidity) under cell culture conditions. After treatment with different concentrations of test compound **1a**, **1b**, **2** or the vehicle DMSO, the cells were incubated for another 24 h (37 °C, 5% CO_2_, 95% humidity). The cells were washed with cytoskeletal buffer (10 mM MES, 3 mM MgCl_2_, 138 mM KCl, 2 mM EGTA, pH 6.8), fixed and permeabilized in 3.7% formaldehyde and 0.2% Triton X-100 in cytoskeletal buffer for 5 min at rt. As additional fixation step, the cells were incubated with ice-cold EtOH for 10 s and rehydrated in PBS. Actin staining was done with Actistain 488 phallodin (100 nM in PBS) for 30 min at rt in the dark. Finally, the cells were washed three times with PBS and the coverslips were embedded in Roti^®^-Mount FluorCare (Roth) with 1 µg/mL DAPI for nuclei staining. Actin filaments and nuclei were documented by confocal microscopy (Leica Confocal TCS SP5, 630× magnification).

#### 2.2.9. Limiting Dilution Assay

Limiting Dilution Assays and analyses using ELDA [24] were performed as described [18]. Briefly, 2048 or 4096 cells were seeded in one row of a 96-well plate and serially diluted using a 1:1 ratio for NCH644 or GS-5 and 17/02 cells, respectively. After seeding, the cells were immediately treated as indicated and after 10 days analyzed for spheres larger than 8 cells. The number of positive wells was counted and entered into ELDA-software to determine the stem-cell frequency, which is defined as the number of cell needed to form a single sphere [24].

#### 2.2.10. Measuring Apoptosis-Induction Using Flow Cytometry

U251 und Mz54 glioblastoma cells (6 × 10^5^ cells/mL, 0.5 mL/well) were seeded in 24-well plates. The next day, the cells were treated with **1a**, **1b** or vehicle (DMSO) for 48 h. Prior to the measurement the cells were harvested and stained using Annexin-V-APC (BD Biosciences, Heidelberg, Germany) and propidium iodide (PI, 0.05 mg/mL, Sigma-Aldrich, Taufkirchen, Germany) as described previously [19]. Cells were analyzed with BD Accuri C6 flow-cytometer (BD Biosciences) and data processing was done using BD Accuri C6 software (BD Biosciences).

## 3. Results

The new title compound **1a** was prepared following a one-pot procedure from the aldol condensation reaction of *N*-methylpiperidin-4-one with two equivalents 4-pentafluorothiobenzaldehyde under basic conditions, in the presence of aqueous NaOH, in methanol at room temperature (Figure 2). **1a** was obtained as a yellow solid in moderate yield and analyzed by NMR spectroscopy, IR spectroscopy and mass spectrometry. The known compounds **1b** and **2** (EF24) were prepared analogously to the synthesis of **1a** following a published procedure [10]. The stability of the substances **1a**, **1b** and **2** in aqueous solution was determined by ^1^H NMR monitoring over 72 h (Appendix A).

### 3.1. Antiproliferative Activity

**1a** was screened for its anti-proliferative activities against nine cancer cell lines from six different tumor entities and one endothelial hybrid cell line (Table 2). The previously published MTT results of the anti-tumorally active curcuminoids EF24 (compound **2**) and **1b** were added for comparison. [10] The results showed that **1a** was more active than **1b** and **2** in all tested cell lines. **1a** showed excellent activities against all cancer lines with IC_50_ values between 0.11 µM and 0.27 µM. The average IC_50_ value of all cancer cell lines (0.19 µM) was slightly lower than that of the endothelial hybrid cell line EA.hy926 (0.31 µM), with a selectivity index (IC_50_ average cancer cells/IC_50_ EA.h926 cells) of 1.6. While the superiority of **1a** over **1b** was only slight in case of the HT-29, HCT-116, DLD-1 and MCF-7/Topo cells, the difference between **1a** and **1b** was greater in the 518A2, KB-V1/Vbl, and Panc-1 cells. The known curcuminoid **2** was up to ten-fold less active than compound **1a** in the tested cancer cell lines. **3** was also reported to show much lower activity than **1a** against HT-29, HCT-116 and Panc-1 cells. [25,26] We also tested **1a** and **1b** on the conventional glioblastoma cell lines U251 and Mz54 because **3** has been reported previously to be effective in GBM cells [24]. Both compounds led to a concentration-dependent decrease of viability after 72 h in both cell lines (Table 2), while their effects after 24 h were less pronounced (data not shown). In the GBM cell lines, the difference between both compounds was very pronounced, with **1a** being up to three times more effective than **1b**. 

The cytotoxicity against colorectal carcinoma (CRC) cells was confirmed in a time dependent hexosaminidase enzyme assay, where **1a** and **1b** strongly inhibited the growth of HCT-116 and SW-480 cells yet after 24 h (Table 3). Again, **1a** was superior in activity (i.e., twice as active) when compared with **1b** and after 72 h both cell lines had responded well to the new compound **1a** which displayed excellent IC_50_ values of up to 74 nM (HCT-116).

### 3.2. Anti-Angiogenic Effect

EF24 (**2**) has been reported to inhibit angiogenesis by reducing VEGF expression levels [27] and secretion [28], so the antiangiogenic properties of test compounds **1a**, **1b** and **2** were evaluated using the in vivo zebrafish embryo assay (Figure 3). [29] Manually dechorionated transgenic *casper*-zebrafish (*fli1a*:GFP) larvae (24 h post fertilization) were treated with non-toxic concentrations of **1a**, **1b**, **2** (10 µM), vehicle (DMSO) or the known angiogenesis inhibitor axitinib (1 µM) as positive control for 48 h. [30] The area of the subintestinal veins (SIV) of at least 20 identically treated zebrafish served as a marker for angiogenic development (Figure 3A). [31] **1a** reduced the SIV area by 47% in contrast to 32% and 15% for **1b** and **2**, compared to vehicle treated embryos set to 100% (Figure 3B). 

### 3.3. Alteration in Cell-Cycle Progression of HT-29 Cells

The influence of compounds **1a**, **1b** and **2** on the cell cycle progression of HT-29 colon carcinoma cells was assessed by PI staining and subsequent flow cytometry (Figure 4A). Representative histograms are shown in the SI (Appendix A). After treatment for 24 h, compound **2** (1 µM) showed an arrest in the G2/M phase and a slight increase in sub-G1 phase cells, whereas the percentage of G1 and S phase cells was decreased when compared to vehicle treated cells. **1a** and **1b** (1 µM) caused no pronounced G2/M arrest but a remarkable increase in sub-G1 cell population, as well as a distinct decrease of G1 and S phase cells (Figure 4A). 

### 3.4. Increase of Intracellular ROS Levels in HT-29 Cells

Reactive oxygen species (ROS) play an important role in the regulation of apoptotic pathways leading to mitochondria-, death receptor- and endoplasmic reticulum-mediated cell death. [32] By increasing endogenous ROS levels above the toxicity threshold, cancer cells can be selectivity killed due to their overall higher ROS concentration compared with normal cells. [33] As previously described, curcuminoids like EF24 (**2**) can affect intracellular ROS levels and thus induce apoptosis in cancer cells. [5,8] For this reason, the intracellular ROS concentration of compounds **1a**, **1b** and **2** was determined by means of the 2´,7´-dichlorohydrofluorescein (DCFH-DA) assay (Figure 4B). [34] At lower concentrations (0.5 µM), **2** showed a stronger ROS increase of 322% compared to 273% for **1a** and 276% for **1b** relative to untreated cells set to 100%. But 2 µM of **1a** and **1b** led to a rapid increase in ROS of up to 408% and 571% respectively, whereas **2** reached ROS levels of 429%. This early and rapid increase in ROS levels, as well as the high proportion of sub-G1 cell population and the activation of effector caspases 3 and 7 (Appendix A), suggest that the induction of apoptosis may be an additional effect of compound **1a**. [35] 

### 3.5. Influence on HT-29 Cell Morphology and Actin Cytoskeleton

After the recognition of morphological changes under the influence of compound **1a**, the effect on the actin cytoskeleton of HT-29 tumor cells was investigated by immunofluorescence imaging (Figure 5). After treatment with **1a** and **1b** for 24 h, we observed a concentration-dependent reorganization of filamentous actin (F-actin) from the characteristic cortical filaments (shown by the untreated control) to cell rounding and actin clustering. (Figure 5A). Most often this results from the release of extracellular matrix (ECM) attachments and reorganization of focal adhesions, leading to a more spherical morphology. [36] While **1a** already showed signs of focal adhesion detachment and cell rounding at concentrations of 1 µM, these effects only became apparent for **1b** at a dose of 2 µM. In contrast, **2** showed neither outstanding change in the actin cytoskeleton nor signs of apoptosis in the same concentration range. In particular, **1a** induced the formation of stress fibers and apoptopodia-like structures, which are characteristic of early stages of apoptotic cells (Figure 5B) [37,38]. We could confirm activation of caspases 3 and 7 through an Apo-One^®^ Homogenous Caspase-3/7 Assay Kit (Appendix A)

### 3.6. Induction of Cell Death in GBM Cell Lines

The induction of early apoptosis and cell death by **1a** and **1b** in U251 and Mz54 GBM cells was evaluated by flow cytometry after staining with Annexin V-APC and PI (Figure 6). Representative dot-plots are shown in Appendix A. In order to better compare the death-promoting potency of the compounds, we selected sub-maximal concentrations derived from the experiments shown in Table 2. Strictly speaking, we treated the cells with 0.3 and 0.6 µM **1a** and 0.5 and 1 µM **1b**. Both compounds led to increased early apoptosis and overall cell death in U251 (Figure 6A) and Mz54 (Figure 6B) tumor cells in a dose- and time-dependent manner. Again, **1a** was active at lower doses when compared with **1b**.

### 3.7. Stemness Decrease of GSCs

Since Curcumin was shown to exert antitumor activity in cancer stem cells [39] and in particular glioma stem-like cells (GSCs) [40], we next tested the compounds using more complex cell lines and employed the GSC sphere cultures GS-5 [17] (Figure 7A,B), NCH644 [18] (Figure 7C,D) and the primary culture 17/02 [19] (Figure 7E,F), which is derived from a second recurrent tumor of a heavily pretreated patient (radiochemotherapy and additional radiotherapy after first remission) and therefore represents an excellent model for treatment-resistant GSCs. In contrast to the conventional cell lines displayed above these cells are cultured under serum-free conditions and are considered a more translational model system since they resemble the original tumor more closely [17]. As an additional positive control we included arsenic trioxide (As_2_O_3_, ATO) which effectively blocks proliferation and stemness in GSCs [19]. Similar to the conventional GBM cell lines we did not observe pronounced growth inhibition after 24 h (data not shown). However after 48 h (Figure 7A,C,E)) and 72 h (Figure 7B,D,F) all three cell lines exhibited concentration-dependent decreases in cell viability with similar IC_50_-values compared to the conventional cell lines (Table 4).

Finally, we tested if the novel compound could also limit the sphere forming potential of the GSCs, which would be indicative of reduced stemness. For this purpose we performed limiting dilution assays (LDA) followed by an analysis with the freely available Web-App ELDA [24] (Figure 8), as described recently [19]. In order to systematically analyze the drug-induced reduction in stemness, we calculated the IC_50_ and IC_25_-values to ensure that their stemness-suppressing activities were not confounded with induction of cell death (Table 4). Accordingly, GS-5 were treated with 50 and 125 nM of **1a**; 360 and 815 nM of **1b**. NCH644 cells received 120 nM and 200 nM **1a** or 525 and 965 nM of **1b**, while the primary culture 17/02 was treated with 80 and 150 nM of **1a** and 400 and 700 nM of **1b**. We analyzed the cells 10 days after seeding and treating. This approach showed that all three GSC cultures, GS-5 (Figure 8A), NCH644 (Figure 8B) and the primary culture 17/02 (Figure 8C) displayed effective and concentration-dependent decreases in their sphere-forming capacities (i.e., stem-cell frequency) by **1b** and in particular **1a**, supporting the notion that this drug is a very promising candidate for further evaluation. The stem-cell frequency is defined as the theoretical amount of cells needed to form a sphere [24] and is summarized in Table 5.

## 4. Discussion

Cancer is one of the leading mortalities worldwide and despite decades of research many cancers, such as glioblastoma, still have devastatingly low survival rates of the afflicted patients exemplifying the need to develop more effective therapeutic agents. Curcumin is the main component of turmeric. Its potent anticancer effects are very well described and is even able to cross the blood-brain barrier [40]. Despite these promising properties, the bioavailability of Curcumin is quite poor, and it is rapidly degraded in vivo [1]. Therefore, novel and improved derivatives have the potential to overcome these drawbacks, while further improving the known anticancer effects. 

Probably the most thoroughly studied representative of the synthetic curcuminoids is (3E,5E)-3,5-bis[(2-fluorophenyl)methylene]-4-piperidinone (EF24), which evokes promising antitumour properties such as cell cycle arrest, apoptosis induction, and inhibition of HIF-α and NF-κB. [41,42,43,44] In our previous work we investigated tetra- and hexa-fluorinated, as well as pentafluorothio-substituted 3,5-bis(benzylidene)-4-piperidinone derivatives such as **1b**. Based on this compound we developed the pentafluorothio-substituted curcuminoid (E)-1-methyl-3,5-bis(4-pentafluorothiobenzylidene)-4-piperidone (**1a**) with superior activity. [11] Due to the triple-digit nanomolar IC50 values against nine different cancer cell lines, **1a** probably belongs to the most active representatives of the synthetic curcuminoids [45]. Comparing the structure with previous studies, both the methyl substituent and the pentafluorothio groups seem to be essential for the increased efficacy of **1a** [11]. 

We could previously demonstrate that Curcumin effectively targets GBM cells both in vitro and in vivo [46,47] and we could further show that this was mediated by inhibition of Signal Transducer and Activator of Transcription (STAT) 3, leading to reduced cell viability, induction of cell death and reduction of cell migration and invasion. Here, we provide evidence that the novel compound **1a** effectively reduces the viability of numerous cancer cell lines derived from multiple cancer entities. Similarly, the lead compound Curcumin (**3**) has been shown to be an effective anticancer agent across several cancer cell lines as well as cancer models in vitro and in vivo. [48] Based on our initial assessment of broad cancer-type agnostic activity we next reasoned that our novel compound **1a** exerts comparable pleiotropic effects by acting on similar processes than Curcumin, albeit at much lower concentrations. As such, we observed potent inhibition of angiogenesis, induction of ROS, cell cycle inhibition and an increase of effector caspases 3 and 7. Similarly, Curcumin has been shown to inhibit angiogenesis for example in models of pancreatic cancer [49] and breast cancer [50]. Furthermore, the angiogenesis assay was performed using live zebrafish embryos using a relatively high concentration (10 µM) and we did not observe any adverse effects, suggesting low in vivo toxicity. The comparison of all tested cancer cell lines with the endothelial hybrid cell line EA.hy926 showed a slight selectivity (selectivity index: 1.6) towards the cancer cells. Although these HUVEC-A549 adenocarcinoma hybrid cells (EA.hy926) are only partially suitable as a healthy comparative cell line, they are significantly less invasive and not capable of forming solid tumors. [51] Additionally, the induction of ROS, increase of the subG1-fraction and activation of effector caspases 3 and 7 and morphological changes of the actin cytoskeleton is likely coupled to induction of cell death via apoptosis in HT29 colon adenocarcinoma. Moreover, we could confirm potent apoptosis-induction using the GBM cell lines.

Importantly, we next focused on a more complex model system by employing three GSC spheroid cell lines, which are known to more closely mimic the original tumor and as such are a better preclinical model system [17]. We could show that **1a** not only decreases viability of GSCs, but also reduces their stem-cell frequency using sub-lethal drug concentrations indicating that **1a** has the potential to target the stem-like phenotype of cancer cells. In fact, it is well-established that GBM cells can obtain a stem-like phenotype in vivo and that these cells are largely responsible for treatment resistance and finally disease recurrence. [52] Although GBM is a prime example for tumors driven by stem-like cancer cells, comparable reports can be found for other cancers. [53] Similarly, numerous reports are available for a targeting of stem-like cancers cells using Curcumin. [39,54] Thus, in summary we were able to synthesize a novel compound with improved anticancer activity and therapeutic potential, which is suitable for further analyses.

## 5. Conclusions

A new Curcumin derivative **1a** was prepared in one step and tested against a panel of tumor cell lines. Our working hypothesis was that **1a** is a superior anticancer drug compared to **1b** and **2** and our primary endpoint of analyses was after assessment of this anticancer activity across several cancer cell lines derived from multiple cancer entities. The compound showed a notable higher anti-proliferative activity than close congeners such as EF24, Curcumin, and **1b**. Cell death and apoptosis induction by **1a** occurred at lower doses than by **1b**. This was further corroborated by a strong increase in ROS production and the Sub-G1 phase of the cell cycle of HT-29 adenocarcinoma by **1a** could be determined. Also, the effect on the actin cytoskeleton in treated cells was clearly improved compared to **1b**. Additionally, **1a** showed similar growth-inhibiting efficacy when tested against several GSC cultures, including the primary culture 17/02, which is derived from a heavily pre-treated GBM patient. **1a** was also able to effectively inhibit stemness in all three cultures in low nanomolar doses. Hence, **1a** appears to be an optimized anti-tumor curcuminoid and further investigation of this promising compound is warranted.

## Figures and Tables

**Figure 1 biomolecules-11-00947-f001:**
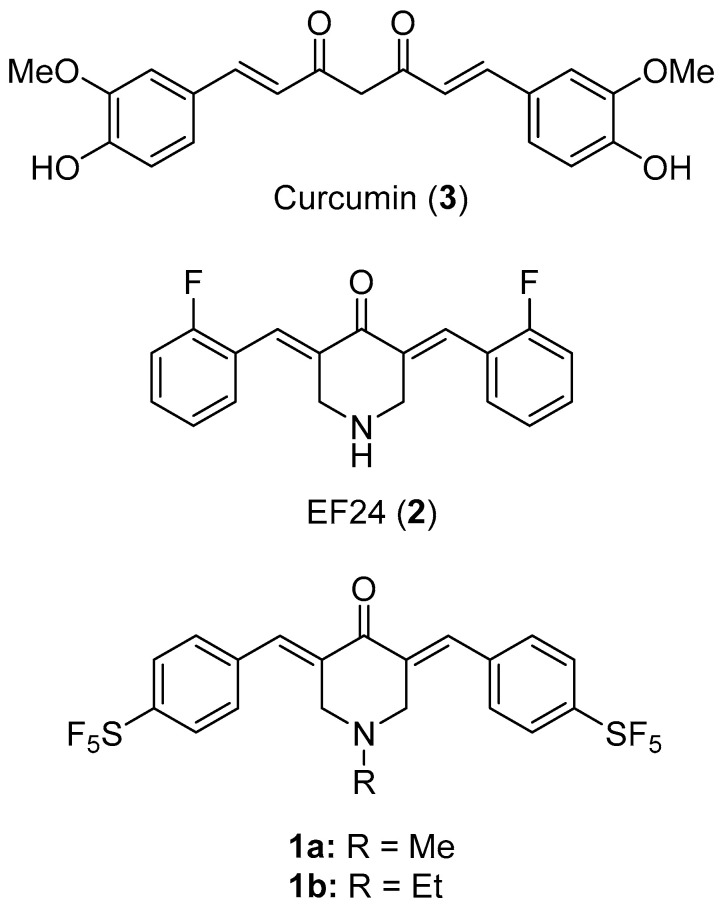
Structures of curcumin (**3**) and the curcuminoids EF24 (**2**), **1a** and **1b**.

**Figure 2 biomolecules-11-00947-f002:**
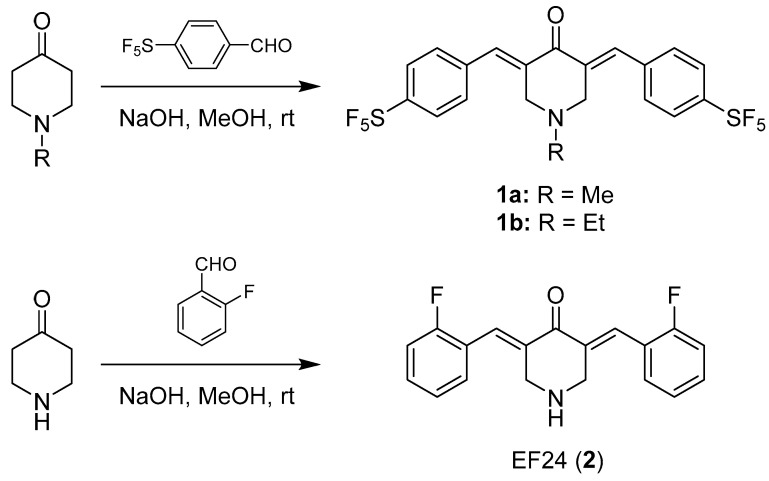
Synthesis of curcuminoids **1a**, **1b** and EF24 (**2**). rt: room temperature.

**Figure 3 biomolecules-11-00947-f003:**
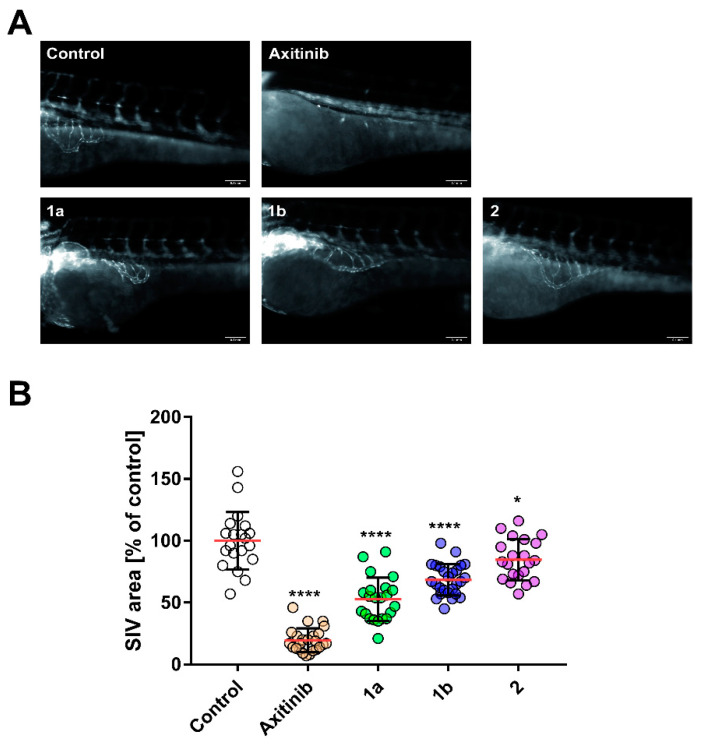
(**A**) Antiangiogenic effect of **1a**, **1b**, **2** (10 µM) or vehicle DMSO in Zebrafish embryos after 48 h exposure. Images are representative of at least 20 identically treated fish. (**B**) The area of sub-intestinal veins (SIV) depicted as the mean ± SD of at least 20 embryos (*n* = 20 per group). *: *p* < 0.05; ****: *p* < 0.0001, One-way ANOVA with Dunnett’s multiple comparison test (GraphPad Prism 7).

**Figure 4 biomolecules-11-00947-f004:**
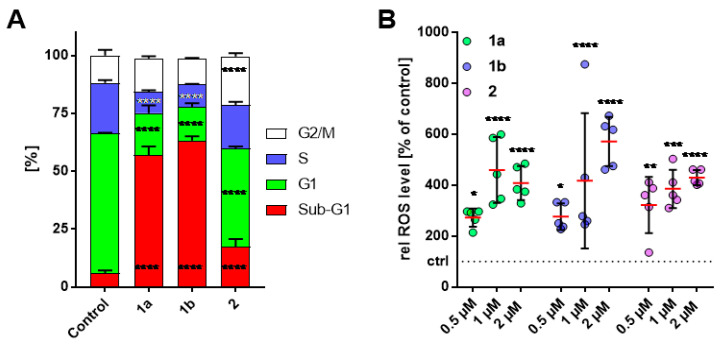
(**A**) Effect of **1a**, **1b** and **2** (1 µM) on the cell-cycle of HT-29 colorectal adenocarcinoma cells after 24h of treatment. Negative controls were treated with an equivalent amount of solvent (DMSO). Box-plot of the quantification (mean ± SD) of the cell cycle distribution of HT-29 cells after treatment with **1a**, **1b** or **2** of three independent assays. The measurements were performed in triplicate and quoted as means ± SD (*n* = 9 per group). (**B**) Levels of reactive oxygen species (ROS) in HT-29 colon adenocarcinoma cells after treatment with **1a**, **1b** and **2** (0.5, 1, 2 µM) for 1 h determined by fluorescence-based DCFH-DA assay. The values are mean values ± SD from at least five independent experiments with negative control treated with an equivalent amount of solvent and set to 100% (dashed line; ctrl) (*n* = 5 per group). *: *p* < 0.05; **: *p* < 0.01; ***: *p* < 0.001; ****: *p* < 0.0001 against control for each cell cycle phase, Two-way ANOVA with Dunnett’s multiple comparison test (GraphPad Prism 7).

**Figure 5 biomolecules-11-00947-f005:**
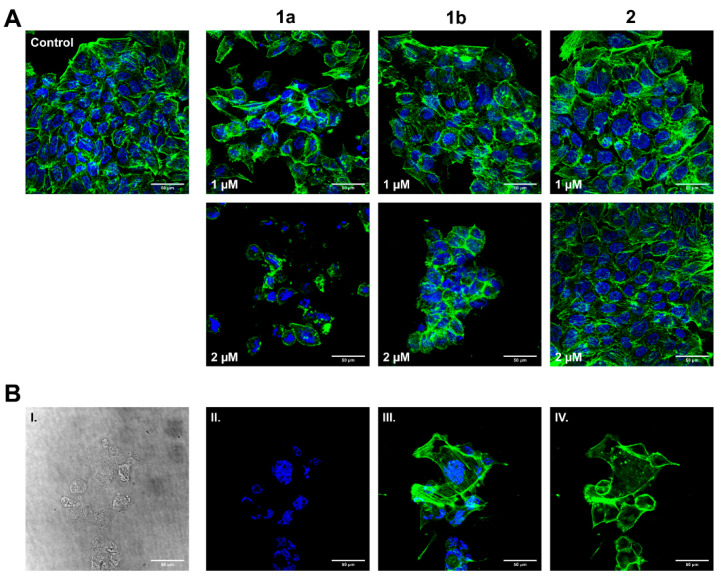
(**A**) Concentration-dependent effect of **1a**, **1b** and **2** (1 and 2 µM) on the actin cytoskeleton of HT-29 colon adenocarcinoma cells. An equivalent amount of solvent (DMSO) served as negative control. Images are representative of at least three independent experiments. Magnification 630× (*n* = 3 per group). (**B**) Substance **1a** (1 µM) causes blebbing (I., bright field), nuclei fragmentation (II., DAPI staining) and different types of actin reorganization (III. and IV., phalloidin staining) such as stress fiber formation, actin clustering, cell rounding and formation of apoptopodia-like structures in HT-29 cells (*n* = 3 per group).

**Figure 6 biomolecules-11-00947-f006:**
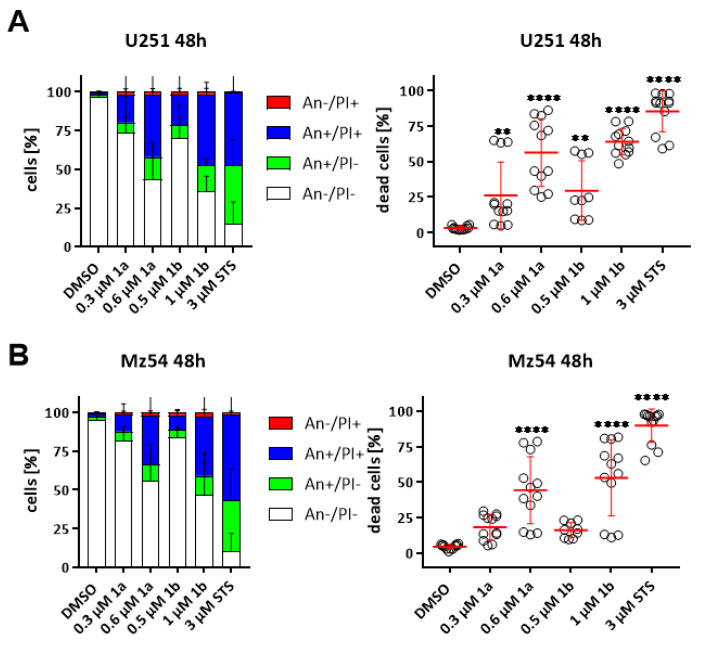
Concentration-dependent induction of cell death in conventional GBM cell lines U251 and Mz54 after treatment for 48 h. FACS-based detection of Annexin V (An)- and Propidium Iodide (PI)-stained cells. (**A**,**B**) Summary displaying (left) the different cell populations of (**A**) U251) and (**B**) Mz54 after treatment as indicated as stacked bar graph with SEM, and (right) the percentage of dead cells (100%—An−/PI−) as a point-plot with the mean (+/− SEM) as a horizontal red line. 3 µM Staurosporine (STS) served as a positive control. (C and D). The data are a summary of at least three experiments performed in triplicates (*n* = 9–12 per group). **: *p* < 0.01; ****: *p* < 0.0001, One-Way ANOVA with Dunnett’s multiple comparison test (GraphPad Prism 7).

**Figure 7 biomolecules-11-00947-f007:**
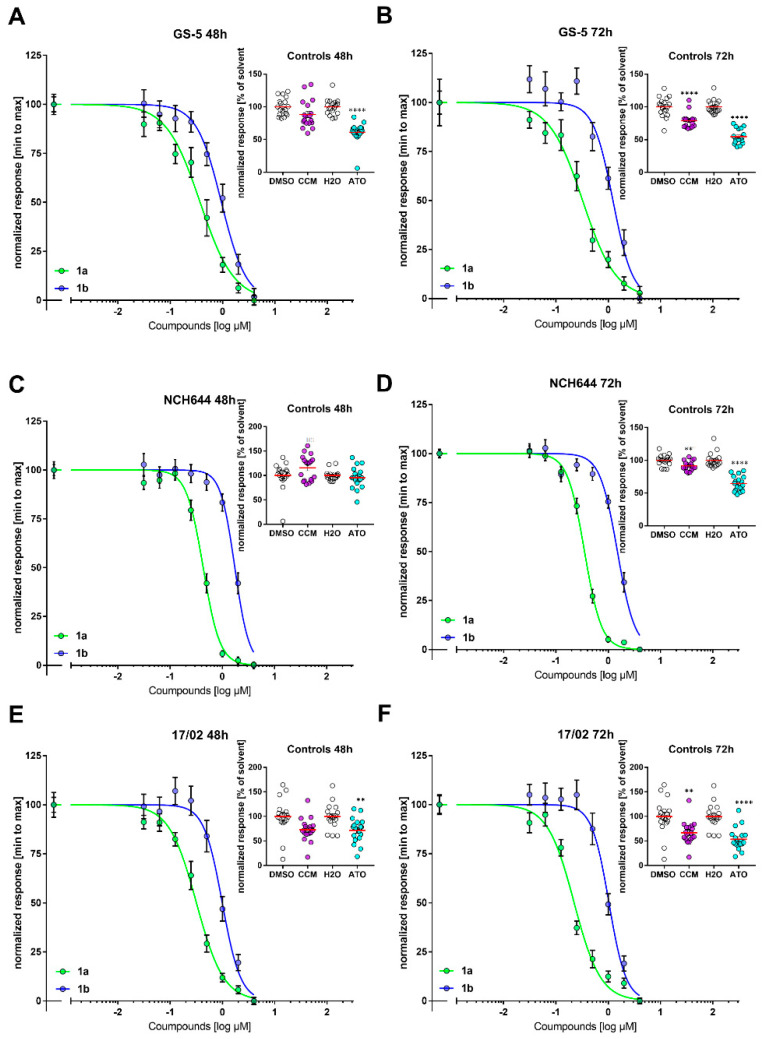
Concentration-dependent decrease in cell viability of the conventional GSC sphere cultures GS-5, NCH644 and the primary culture 17/02. MTT-assay of (**A**,**B**) GS-5, (**C**,**D**) NCH644 and (**E**,**F**) 17/02 GSCs treated for (**A**,**C**,**E**) 48 h or (**B**,**D**,**F**) 72 h with compounds as indicated. The data were normalized from highest (100%) to lowest (0%) and plotted against the log-dose of the compounds. To calculate the IC50-values (Table 4) a curve-fitting was done using the function “log(inhibitor) vs. normalized response Variable slope” without constraints (GraphPad Prism 7). As positive controls, the cells were treated with 20 µM Curcumin (CCM) or solvent (DMSO) or 2.5 µM arsenic trioxide (As_2_O_3_, ATO) or solvent (H_2_O) for the same amount of time (Inserts in A to D). The data presented is the summary of three independent experiments performed in 6 replicates (*n* = 18 per group). **: *p* < 0.01; ****: *p* < 0.0001; Unpaired t test with Welch’s Correction (GraphPad Prism 7).

**Figure 8 biomolecules-11-00947-f008:**
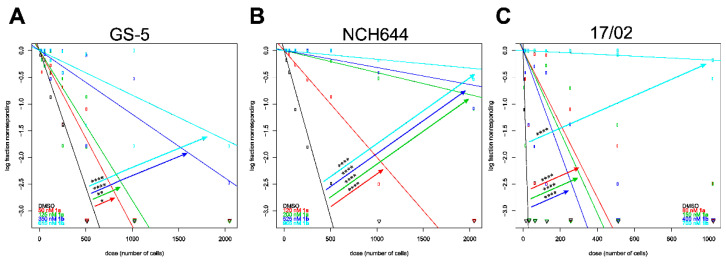
Reduced stemness of the GSC cultures GS-5, NCH644 and the primary culture 17/02. Log-fraction plot of the limited dilution model of data from (**A**) GS-5, (**B**) NCH644 GSCs and (**C**) the primary culture 17/02 treated with solvent (DMSO, black), the IC_25_ (red) and IC_50_ (green) of **1a** and the IC_25_ (dark blue) or IC_50_ (light blue) of **1b** for 10 days and analyzed using ELDA software [18]. The data are the summary of 2 experiments performed in 12 replicates (*n* = 24 per group). *: *p* < 0.05; **: *p* < 0.01; ****: *p* < 0.0001; Pairwise Test for Differences using Chi-square-test (ELDA). The estimated stem-cell frequency (number of cells needed to form a sphere) is given in Table 3.

**Table 1 biomolecules-11-00947-t001:** Overview of the analyzed cell lines and their tumor type.

Cell line	Tumor Type	Reference
518A2	Melanoma	[14]
Panc-1	Pancreatic ductular adenocarcinoma	ACC-783
KB-V1^Vbl^	Cervix carcinoma, Vinblastine resistant	ACC-149
MCF-7^Topo^	Breast Carcinoma, Topotecan resistant	ACC-115
HT-29	Colon Carcinoma	ACC-299
HCT-116	Colon Carcinoma	ACC-581
DLD-1	Colon Carcinoma	ACC-278
EA.hy926	Endothelial hybrid	CRL-2922
SW-480	Colon Carcinoma	CCL-228
U251	Glioblastoma	ECACC 09063001
Mz54	Glioblastoma	CVCL_M406
GS-5	Glioblastoma, stem-like cells	[17]
NCH644	Glioblastoma, stem-like cells	[18]
17/02	Glioblastoma, stem-like cells	[19]

**Table 2 biomolecules-11-00947-t002:** Inhibitory concentrations IC_50_
^[a]^ [µM] of **1a**, **1b**
^[b]^, EF24 (**2**) ^[b]^ and Curcumin (**3**) ^[c]^ when applied to cells of human 518A2 melanoma, KB-V1^Vbl^ cervix carcinoma, Panc-1 pancreatic ductular adenocarcinoma, MCF-7^Topo^ breast carcinoma, HT-29, HCT-116 and DLD-1 colon (adeno-)carcinomas, EA.hy926 endothelial hybrid cells and U251 and Mz54 glioblastoma.

Compounds	1a	1b ^[b]^	EF24 (2) ^[b]^	Curcumin (3)
**518A2**	0.17 ± 0.02	0.99 ± 0.14	1.8 ± 0.2	-
**HT-29**	0.20 ± 0.01	0.29 ± 0.02	1.6 ± 0.1	13.3 ^[c]^
**HCT-116**	0.11 ± 0.01	0.24 ± 0.02	1.5 ± 0.2	10.9 ^[c]^
**DLD-1**	0.16 ± 0.01	0.33 ± 0.01	1.3 ± 0.1	-
**KB-V1^Vbl^**	0.15 ± 0.01	0.64 ± 0.05	1.1 ± 0.1	-
**MCF-7^Topo^**	0.22 ± 0.02	0.40 ± 0.01	2.2 ± 0.2	-
**Panc-1**	0.19 ± 0.01	1.1 ± 0.1	1.5 ± 0.2	20.4 ^[c]^
**EA.hy926**	0.31 ± 0.03	0.54 ± 0.03	1.4 ± 0.1	
**U251**	0.22 ± 0.02	0.79 ± 0.03	-	-
**Mz54**	0.27 ± 0.02	0.83 ± 0.02	-	-

^[a]^ Values are the means ± SD (standard deviation) of four independent experiments. They were derived from concentration–response curves obtained by measuring the percentage of vital cells relative to untreated controls after 72 h using MTT-assay. ^[b]^ Values of EF24 (**2**) and **1b** (except for Mz54 and U251 data) were taken from Schmitt et al. [10] ^[c]^ Values for curcumin were taken from Cen et al. (HT-29 and HCT-116 cells), and Friedman et al. (Panc-1 cells) [25,26].

**Table 3 biomolecules-11-00947-t003:** Inhibitory concentrations IC_50_ [µM] of **1a** and **1b** when applied to human HCT-116 and SW-480 colon carcinoma cells ^[a]^.

		HCT-116			SW-480	
	24 h	48 h	72 h	24 h	48 h	72 h
**1a**	0.16 ± 0.02	0.073 ± 0.01	0.074 ± 0.01	0.56 ± 0,1	0.11 ± 0.02	0.1 ± 0.01
**1b**	0.34 ± 0.04	0.14 ± 0.02	0.15 ± 0.01	0.75 ± 0.14	0.22 ± 0.04	0.17 ± 0.03
**2**	0.47 ± 0.07	0.26 ± 0.07	0.29 ± 0.03	1.1 ± 0.3	0.37 ± 0.07	0.24 ± 0.03

^[a]^ Values are the means ± SD (standard deviation) of four independent experiments. They were derived from concentration–response curves obtained by measuring the percentage of vital cells relative to untreated controls after 24, 48 and 72 h using a hexoaminidase-assay.

**Table 4 biomolecules-11-00947-t004:** Inhibitory concentrations IC_50_ [µM] of **1a**, **1b** when applied to the GSCs GS-5, NCH644 and the primary culture 17/02 (*n* = 18 per group).

Compound	1a		1b	
IC_50_ after Incubation	48 h	72 h	48 h	72 h
**GS-5**	0.37 ± 0.04	0.32 ± 0.05	0.95 ± 0.04	1.3 ± 0.04
**NCH644**	0.43 ± 0.02	0.35 ± 0.01	1.7 ± 0.02	1.5 ± 0.02
**17/02**	0.32 ± 0.03	0.22 ± 0.03	0.99 ± 0.04	1.0 ± 0.03

**Table 5 biomolecules-11-00947-t005:** The estimated stem-cell frequency (1/X) of the GSCs GS-5, NCH644 and the primary culture 17/02 after treatment with solvent (DMSO) or **1a** or **1b** with the IC25 and IC50 (*n* = 24 per group).

Compound	DMSO	1a		1b	
Stem-Cell Frequency (1/X)		IC_25_	IC_50_	IC_25_	IC_50_
**GS-5**	196	305	357	839	1199
**NCH644**	8	145	131	104	5626
**17/02**	216	1680	4288	5169	11,319

## Data Availability

Raw data can be made available upon reasonable request. No large-scale datasets have been generated or used.

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
