# Peer review of "A New Pentafluorothio-Substituted Curcuminoid with Superior Antitumor Activity"

_biomolecules, 2021, doi:10.3390/biom11070947_

Round 1

Reviewer 1 Report

In this paper Linder et al. investigated the in vitro anti-tumoral activity of a new pentafluorothio-substituted curcuminoid in a large panel of cancer cell lines, and especially on GBM cells.

The introduction is appropriate and to the point, with sufficient references.

However, in the opinion of the Reviewer, there are some critical aspects both in the Material and Methods section and in the Results which need to be revised, as reported below.

Abstract

- Line 23: “…was superior to the known anti-tumoral active..”

- Lines 26-27: “A considerable activation of the executioner caspase-3 and -7..”

- Line 28: “membrane protrusion suggest the triggering..”

- Line 29: “the re-organization of the actin cytoskeleton..”

- Line 29: “the formation of stress fibers..”

- Lines 30-32: Please rewrite this sentence, since it is too long and verbose.

Introduction

- Lines 38 and 43: “Curcumin (3)” and “EF24 (2)”; what does the number in brackets stand for? Please clarify.

- Lines 43, 46 and 54: “bis-benzylidene-piperidone” or “bis(benzylidene)piperidone”?

- Lines 47-49: This is important and needs further elaboration and references.

Material and Methods

- Paragraph 2.1.2: Please revise this paragraph since there are few unclear symbols (see lines 73, 75, 77).

- Paragraph 2.2.1: there are a lot of tumor cell lines being tested; maybe a table would help the reader, with possibly a distinction between for instance GBM cell lines and other cancer cell lines.

- Line 108: non-GBM cells; please correct.

- Paragraph 2.2.3. Hexoseaminidase enzyme assay. On which cell lines this test was performed? Please specify.

- Line 152: HT29 or HT-29? please correct through the text.

Line 193: 2048 (NCH644) or 4096 (GS-5 and 17/02) cells were seeded? 2048 or 4096 refer to the number of seeded cells?

Results

- Line 216: this should be Figure 2. Please correct the figure legend, since it is the same as figure 1.

- Line 219: (Tbl. 1). Please correct. The same for all the other Tables mentioned in the text.

- Paragraph 3.1: 3.1 Anti-proliferative

- Paragraph 3.1. Why the anti-proliferative activity was performed only on 9 cancer cell line? Why SW-480 and the glioma stem-like cells were excluded from this analysis?

- Lines 243-247: why the hexosaminidase enzyme assay was not performed on DLD-1 colon carcinoma cells?

- Line 245: Table 2. The same as Table 1.

- Line 255: Paragraph 3.1. Anti-angigenic effect  please correct: “3.2. Anti-angiogenic effect”

(please correct all the subsections of the Result section, since they are all reported as 3.1)

- Line 263: Please correct into Figure 3B.

- Figure 3. Maybe this figure should be divided into two different figures, since panel A and B report the results concerning the anti-angiogenic effects of the compounds, whereas panel C and D refer to DCFH-DA assay and cell cycle analysis on HT-29 colorectal adenocarcinoma cells. Please change the figure legend accordingly.

- Line 280. 3.1. Increase of intracellular ROS levels in HT29 cells. Please correct

- Why DCFH-DA assay, cell-cycle analysis and immunofluorescence staining of actin cytoskeleton were performed only on HT-29 colorectal adenocarcinoma cells? why not on the other colon carcinoma cell lines or in all other tumor cancer cell lines?

- Figure 3D: The Authors should report representative flow cytometry profiles.

- Line 302: the same as before

- Lines 310-315: The Authors state that 1a induced early sings of apoptosis in HT-29 cells. However, the authors need to perform some additional assays e.g. at molecular level by the Western blot analysis of apoptotic marker such as PARP cleavage, caspases such as Caspase-3 and -9 to determine whether the compounds 1a, 1b and 2-induced cytotoxicity is mediated via apoptotic-or non-apoptotic mechanism.

- Line 324: the same as before

- Line 325 and 330: U251 or U251-MG? MZ54 or Mz-54? Please correct through the text.

- Line 326: flow cytometry after staining with Annexin V-APC and PI. This should be reported in the Material and Methods section.

- Figure 5: The Authors should report representative flow cytometry profiles.

- Figure 5: For each figure, the subheading of a, b, c, d, etc., should be mentioned in the text.

- Line 343: the same as before

- Figures 6 and 7: For each figure, the subheading of a, b, c, d, etc., should be mentioned in the text.

- The Authors should perform additional morphological/functional assay to determine whether the novel compound reduces GSC stemness e.g. by analyzing the expression of cancer stem cell markers through RNA sequencing, immunocytochemistry, WB or others.

Discussion and Conclusions:

The English language and style are fine. However, there are a few places with wording that needs to be smoothed over. Minor point: there are some grammatical/spelling errors and minor spell check is required.

Author Response

Please find our replys below each point:

Abstract

- Line 23: “…was superior to the known anti-tumoral active..”

- Lines 26-27: “A considerable activation of the executioner caspase-3 and -7..”

- Line 28: “membrane protrusion suggest the triggering..”

- Line 29: “the re-organization of the actin cytoskeleton..”

- Line 29: “the formation of stress fibers..”

- Lines 30-32: Please rewrite this sentence, since it is too long and verbose.

Answer: Thank you for the careful reading of our manuscript and pointing out these grammatical mistakes. We addressed them accordingly.

Introduction

- Lines 38 and 43: “Curcumin (3)” and “EF24 (2)”; what does the number in brackets stand for? Please clarify.

Answer: “Curcumin” and “3”, as well as “EF24” and “2” are synonyms for the same drug, respectively. The number refers the applied nomenclature for these drugs in the later figures (Fig. 3ff). For clarification, we re-introduced to numbering at the respective part in the results again.

- Lines 43, 46 and 54: “bis-benzylidene-piperidone” or “bis(benzylidene)piperidone”?

Answer: Corrected.

- Lines 47-49: This is important and needs further elaboration and references.

Answer: We thank the referee for this comment. The following sentence was added to the introduction: ´´The SF5 group is a chemically stable xenobiotic mimic of negatively charged biomolecules and it was labelled as ´´super trifluoromethyl group´´, which is often superior to more common fluoro and trifluoromethyl substituents in bioactive compounds [12].´´

Material and Methods

- Paragraph 2.1.2: Please revise this paragraph since there are few unclear symbols (see lines 73, 75, 77).

Answer: The unclear symbols were corrected.

- Paragraph 2.2.1: there are a lot of tumor cell lines being tested; maybe a table would help the reader, with possibly a distinction between for instance GBM cell lines and other cancer cell lines.

Answer: Thank you for pointing this out. We now include a table of all cell line and their tumor-of-origin in the materials and methods section.

- Line 108: non-GBM cells; please correct.

Answer: corrected

- Paragraph 2.2.3. Hexoseaminidase enzyme assay. On which cell lines this test was performed? Please specify.

Answer: HCT-116 and SW-480 cells were applied for this assay. The cell lines are mentioned in this section in the revised version of the manuscript now.

- Line 152: HT29 or HT-29? please correct through the text.

Answer: Done.

Line 193: 2048 (NCH644) or 4096 (GS-5 and 17/02) cells were seeded? 2048 or 4096 refer to the number of seeded cells?

Answer: Thank you again for your careful proof-reading. We have corrected accordingly.

Results

- Line 216: this should be Figure 2. Please correct the figure legend, since it is the same as figure 1.

- Line 219: (Tbl. 1). Please correct. The same for all the other Tables mentioned in the text.

- Paragraph 3.1: 3.1 Anti-proliferative

Answer: We have corrected these mistakes accordingly.

- Paragraph 3.1. Why the anti-proliferative activity was performed only on 9 cancer cell line? Why SW-480 and the glioma stem-like cells were excluded from this analysis?

Our intention was to apply our novel compound to a broad set of cancer cell lines of different tumor entities, but this was meant  only to give a rough general overview of the spectrum of activity, so only the cell lines shown were tested.

The data on the hexoseaminidase assay was provided by our collaboration partner from Kansas University (Anant group) and was intended to highlight the time-dependent antiproliferative effect using an additional assay.  Since our laboratories in Bayreuth and Frankfurt do not have the same SW-480 cell line, the MTT assay could not be done for the SW-480 cells. Hoewever, we would like to point out that the IC50 values of 1a in HCT-116 cells from the MTT assay and the hexoseaminidase assay are very similar. Thus, we are confident that any MTT assay of 1a with SW-480 cells would also provide very similar results to the given hexosaminadase assay results and hope the reviewer agrees  that an additional MTT assay with SW-480 cells is not mandatory.

We decided to separate the data from stem-like cells in the mansucript, because these represent a more complex model and we wanted to make this distinction between “conventional” and “complex” cell lines clearer. We have now added an additional sentence emphasizing this distinction.

- Lines 243-247: why the hexosaminidase enzyme assay was not performed on DLD-1 colon carcinoma cells?

Anwer: Since we could already show (Tbl. 2) that the effects on cell viability/proliferation of 1a are largely cell-type-agnostic, we reasoned that it is sufficient to perform an additional toxicity assay (Tbl. 3) using “only” two cell lines.

- Line 245: Table 2. The same as Table 1.

- Line 255: Paragraph 3.1. Anti-angigenic effect à please correct: “3.2. Anti-angiogenic effect”

(please correct all the subsections of the Result section, since they are all reported as 3.1)

- Line 263: Please correct into Figure 3B.

Answer: We have corrected these typos accordingly

- Figure 3. Maybe this figure should be divided into two different figures, since panel A and B report the results concerning the anti-angiogenic effects of the compounds, whereas panel C and D refer to DCFH-DA assay and cell cycle analysis on HT-29 colorectal adenocarcinoma cells. Please change the figure legend accordingly.

Answer: Excellent suggestion. To avoid white space in our figures we also switched the cell cycle analyses (now paragraph 3.3) and ROS analyses (paragraph 3.4).

- Line 280. 3.1. Increase of intracellular ROS levels in HT29 cells. Please correct

Answer: corrected.

- Why DCFH-DA assay, cell-cycle analysis and immunofluorescence staining of actin cytoskeleton were performed only on HT-29 colorectal adenocarcinoma cells? why not on the other colon carcinoma cell lines or in all other tumor cancer cell lines?

Answer: Our intention was to provide proof-of-concept data showing the biological activity of our novel compound. We have shown broad anticancer activity using the data provided in table 1, indicating that 1a has general anticancer activity. For the more directed approaches, we decided to focus on only one cell line but to employ a variety of assays to show that 1a targets comparable pathways than curcumin or EF24, but with higher efficiency. We have added an additional sentence into the discussion section for clarification.

- Figure 3D: The Authors should report representative flow cytometry profiles.

Answer: We have included representative histograms.

- Line 302: the same as before

Answer: corrected

- Lines 310-315: The Authors state that 1a induced early sings of apoptosis in HT-29 cells. However, the authors need to perform some additional assays e.g. at molecular level by the Western blot analysis of apoptotic marker such as PARP cleavage, caspases such as Caspase-3 and -9 to determine whether the compounds 1a, 1b and 2-induced cytotoxicity is mediated via apoptotic-or non-apoptotic mechanism.

Answer: We performed an additional assay to show an increase in effector caspases 3 and 7 induced by 1a in HT-29 colon carcinoma cells and changed the sentence where we stated a possible apoptosis induction.

- Line 324: the same as before

Answer: corrected

- Line 325 and 330: U251 or U251-MG? MZ54 or Mz-54? Please correct through the text.

Answer: Thank you for bringing this to our attention. It is now corrected.

- Line 326: flow cytometry after staining with Annexin V-APC and PI. This should be reported in the Material and Methods section.

Answer: This method was already stated in the materials and methods section, although we have to acknowledge that the previous title of section 2.2.9 was somewhat misleading. We have now re-phrased it.

- Figure 5: The Authors should report representative flow cytometry profiles.

Answer: We do now provide representative dot-plots as supplementary figures in the Supporting Information.

- Figure 5: For each figure, the subheading of a, b, c, d, etc., should be mentioned in the text.

Answer: We have carefully proof-read the text and made sure that each sub-figure is mentioned in the text.

- Line 343: the same as before

Answer: corrected

- Figures 6 and 7: For each figure, the subheading of a, b, c, d, etc., should be mentioned in the text.

Answer: corrected

- The Authors should perform additional morphological/functional assay to determine whether the novel compound reduces GSC stemness e.g. by analyzing the expression of cancer stem cell markers through RNA sequencing, immunocytochemistry, WB or others.

Answer: Thank you for this suggestion. Due to time restrictions, we were unfortunately not able to perform these additional control experiments. However we would like to point out that the limiting dilution assay and especially its quantitative analyses using ELDA [1] is a very widely accepted approach to measure stemness in vitro. In fact, this tool has been cited over 800 times already and the effects found using this functional assay usually relate to changes in stemness markers as we could show previously [2]. Additionally, the base-compound curcumin has already been shown to inhibit stemness including reduction in marker expression in numerous studies [3-9]. 

Discussion and Conclusions:

The English language and style are fine. However, there are a few places with wording that needs to be smoothed over. Minor point: there are some grammatical/spelling errors and minor spell check is required.

Answer: Thank you for valuable input. We have carefully re-assessed our discussion section and changed accordingly.

References

  1. Hu, Y.; Smyth, G.K. ELDA: extreme limiting dilution analysis for comparing depleted and enriched populations in stem cell and other assays. J Immunol Methods 2009, 347, 70-78, doi:10.1016/j.jim.2009.06.008.
  2. Linder, B.; Wehle, A.; Hehlgans, S.; Bonn, F.; Dikic, I.; Rodel, F.; Seifert, V.; Kogel, D. Arsenic Trioxide and (-)-Gossypol Synergistically Target Glioma Stem-Like Cells via Inhibition of Hedgehog and Notch Signaling. Cancers (Basel) 2019, 11, doi:10.3390/cancers11030350.
  3. Hu, C.; Li, M.; Guo, T.; Wang, S.; Huang, W.; Yang, K.; Liao, Z.; Wang, J.; Zhang, F.; Wang, H. Anti-metastasis activity of curcumin against breast cancer via the inhibition of stem cell-like properties and EMT. Phytomedicine 2019, 58, 152740, doi:10.1016/j.phymed.2018.11.001.
  4. Sordillo, L.A.; Sordillo, P.P.; Helson, L. Curcumin for the Treatment of Glioblastoma. Anticancer Res 2015, 35, 6373-6378.
  5. Sordillo, P.P.; Helson, L. Curcumin and cancer stem cells: curcumin has asymmetrical effects on cancer and normal stem cells. Anticancer Res 2015, 35, 599-614.
  6. Zendehdel, E.; Abdollahi, E.; Momtazi-Borojeni, A.A.; Korani, M.; Alavizadeh, S.H.; Sahebkar, A. The molecular mechanisms of curcumin's inhibitory effects on cancer stem cells. J Cell Biochem 2019, 120, 4739-4747, doi:10.1002/jcb.27757.
  7. Ramasamy, T.S.; Ayob, A.Z.; Myint, H.H.; Thiagarajah, S.; Amini, F. Targeting colorectal cancer stem cells using curcumin and curcumin analogues: insights into the mechanism of the therapeutic efficacy. Cancer Cell Int 2015, 15, 96, doi:10.1186/s12935-015-0241-x.
  8. Wang, D.; Kong, X.; Li, Y.; Qian, W.; Ma, J.; Wang, D.; Yu, D.; Zhong, C. Curcumin inhibits bladder cancer stem cells by suppressing Sonic Hedgehog pathway. Biochem Biophys Res Commun 2017, 493, 521-527, doi:10.1016/j.bbrc.2017.08.158.
  9. Wu, L.; Guo, L.; Liang, Y.; Liu, X.; Jiang, L.; Wang, L. Curcumin suppresses stem-like traits of lung cancer cells via inhibiting the JAK2/STAT3 signaling pathway. Oncol Rep 2015, 34, 3311-3317, doi:10.3892/or.2015.4279.

Reviewer 2 Report

This manuscript is based on the preparation and anti-proliferative, pro-apoptotic and cancer stem cell-differentiating effects of pentafluorothiophenyl-substituted N-methyl-piperidone curcuminoid. Compound 1a does have superior in vitro anti-proliferative activity as judged by the IC50 values in the nanomolar concentration range, relative to the curcuminoid EF24 (2) and its N-ethyl-piperidone analog 1b.  However, there are concerns that the concentration of 1a used in various experiments, including those that evaluate the pro-apoptotic and cancer stem cell-differentiating effects, are much higher than those shown to be effective in Tables 1 or 2. Importantly, the inclusion of the cell lines to study the effects of the compounds lacks clear rationale. Multiple normal cells are not included in the study. The primary target of this compound is not known and more focused experiments in appropriate cell culture and mouse model systems are required to evaluate the therapeutic relevance of this new compound in order to demonstrate that it is indeed superior in vivo to curcumin, which has been extensively studied.

Specific comments.

  1. The NMR data should be shown for all the compounds synthesized.
  2. Figure 3 (antiangiogenic effect) uses of 10 μm and Figure 4 (actin cytoskeleton, apoptosis, etc) uses 1 to 2 μm amounts of the compound 1a that are much higher than the IC50 amounts required for growth inhibition in Table 1. The experiments shown in Figures 3 and 4 should include IC50 concentrations in order to be able to compare between the data in various figures. It is not clear why curcumin is not consistently used for comparison in all the experiments.
  3. The inclusion of cancer cell lines in various experiments lacks clear rationale. The authors should choose a panel of drug-sensitive and ideally genetically-matched drug-resistant cell lines, focusing on specific types of cancers- breast, prostate, colon, etc., and those with a panel of known mutations such as lung cancer.
  4. Multiple normal cells should be tested for the anti-proliferative and pro-apoptotic effects of the compounds.
  5. Is 1a stable in vivo (for example, in mice). Is it absorbed and bioavailable following oral gavage.
  6. Does 1a inhibit tumor growth in mouse models.

Author Response

Please find our replys below each point:

Specific comments.

1. The NMR data should be shown for all the compounds synthesized.

Answer: We have prepared an SI for the NMR data and other results. In addition, the hydrolysis stability of the test substances was also checked by the addition of D2O within 72 h by means of NMR.

2. Figure 3 (antiangiogenic effect) uses of 10 μm and Figure 4 (actin cytoskeleton, apoptosis, etc) uses 1 to 2 μm amounts of the compound 1a that are much higher than the IC50 amounts required for growth inhibition in Table 1. The experiments shown in Figures 3 and 4 should include IC50 concentrations in order to be able to compare between the data in various figures. It is not clear why curcumin is not consistently used for comparison in all the experiments.

Answer: Thank you for this excellent remark. For the functional cell-based assay we used concentrations above the IC50-value determined using the proliferation/viability assay, because based on our previous experience we know that IC50-values derived from proliferation/viability assays (MTT-assays) cannot be transferred 1:1 to these other assays. Especially from a biological point-of-view it does make perfect sense that inhibition of proliferation normally requires lower doses of any drug than to induce cell death. This is probably best exemplified with the conventional GBM cell lines U251 and Mz54 in our manuscript. Accordingly, these cells have an IC50 for proliferation inhibition (Tbl. 2) of 0.22 and 0.27 µM after 72h of treatment with 1a respectively. In contrast, 0.3 µM 1a only induces ~25% cell death after 48h and 0.6 µM approximately ~50%. This comparison nicely shows that some adjustments in concentrations need to be made for the different functional readouts. Nonetheless, this compound requires very favourable drug concentrations for targeting of cancer cells.

We would also like to point out, that for the analyses of stemness we explicitely made sure to apply sub-lethal concentrations of the drugs and used concentrations that represent the IC25 and IC50 of either drug. As such, the GSCs are treated with concentrations of 1a ranging from 50 nM (GS-5, IC25) to a maximum of 200 nM (NCH644, IC50). We have now added the exact concentrations applied in the respective section for clarification.

The large differences in the substance concentrations used are due to the different model systems (zebrafish assay) as well as to much shorter incubation times (DCFH-DA assay, cell cycle analysis, actin cytoskeleton and apoptosis). The drug concentrations for the zebrafish assay admittedly are relatively high but were necessary for the observation of significant anti-angiogenic effects. Nonetheless, would we like to point out that we did not observe any adverse events related to the treatment indicating that the compounds do not induce considerable non-specific toxicities. We have now added an additional discussion concerning this issue. The IC50 values determined by the MTT assay refer to an incubation time of 72 hours and are difficult to transfer to other assays with much shorter treatment times. After several trials, we decided to use concentrations of about 1 µM for optimal results.

3. The inclusion of cancer cell lines in various experiments lacks clear rationale. The authors should choose a panel of drug-sensitive and ideally genetically-matched drug-resistant cell lines, focusing on specific types of cancers- breast, prostate, colon, etc., and those with a panel of known mutations such as lung cancer.

Answer: Thank you again for this delighting remark. Our intention was to apply our novel compound to a broad set of cancer cell lines, since curcumin has already been widely applied to numerous cancer models [1-7], including models of drug resistance. Thus our broad cell panel. As also suggested by the reviewer we already placed a focus on GBM cells including GSCs at the end of the results part. We included at several sections additional sentences highlighting our rationale to increase to clarity of our approach in the present study. Your excellent suggestion will allow us to further elucidate the therapeutic potential of our drugs in drug-sensitive vs -resistant cancer models in follow-up studies.

4. Multiple normal cells should be tested for the anti-proliferative and pro-apoptotic effects of the compounds.

Answer: Thank you for pointing this out. As mentioned above we can show that our compounds are non-toxic in vivo for concentrations up to 10 µM using the zebrafish assay, which are three magnitudes higher than the IC50 values against the tested cancer cell lines. We have now emphasized these findings in the discussion. We have additionally tested our substances on an endothelial hybrid cell line (EA.hy926; HUVEC-A549 adenocarcinoma hybrid), which possess many properties of "healthy" cells. [8] As described in the results, the test compound 1a showed a slight selectivity against cancer cells, compared to the EA.hy926 endothelial hybrid cells tested.

5. Is 1a stable in vivo (for example, in mice). Is it absorbed and bioavailable following oral gavage.

6. Does 1a inhibit tumor growth in mouse models.

Answer: Since both questions are related we chose to answer them together. These are excellent questions, and we can state, as above, that 1a is non-toxic in zebrafish. We are also aware from additional collaboration partner working on melanoma that it inhibits tumor growth in mice. However, these promising data are not yet published.

References

  1. Hu, C.; Li, M.; Guo, T.; Wang, S.; Huang, W.; Yang, K.; Liao, Z.; Wang, J.; Zhang, F.; Wang, H. Anti-metastasis activity of curcumin against breast cancer via the inhibition of stem cell-like properties and EMT. Phytomedicine 2019, 58, 152740, doi:10.1016/j.phymed.2018.11.001.
  2. Sordillo, L.A.; Sordillo, P.P.; Helson, L. Curcumin for the Treatment of Glioblastoma. Anticancer Res 2015, 35, 6373-6378.
  3. Sordillo, P.P.; Helson, L. Curcumin and cancer stem cells: curcumin has asymmetrical effects on cancer and normal stem cells. Anticancer Res 2015, 35, 599-614.
  4. Zendehdel, E.; Abdollahi, E.; Momtazi-Borojeni, A.A.; Korani, M.; Alavizadeh, S.H.; Sahebkar, A. The molecular mechanisms of curcumin's inhibitory effects on cancer stem cells. J Cell Biochem 2019, 120, 4739-4747, doi:10.1002/jcb.27757.
  5. Ramasamy, T.S.; Ayob, A.Z.; Myint, H.H.; Thiagarajah, S.; Amini, F. Targeting colorectal cancer stem cells using curcumin and curcumin analogues: insights into the mechanism of the therapeutic efficacy. Cancer Cell Int 2015, 15, 96, doi:10.1186/s12935-015-0241-x.
  6. Wang, D.; Kong, X.; Li, Y.; Qian, W.; Ma, J.; Wang, D.; Yu, D.; Zhong, C. Curcumin inhibits bladder cancer stem cells by suppressing Sonic Hedgehog pathway. Biochem Biophys Res Commun 2017, 493, 521-527, doi:10.1016/j.bbrc.2017.08.158.
  7. Wu, L.; Guo, L.; Liang, Y.; Liu, X.; Jiang, L.; Wang, L. Curcumin suppresses stem-like traits of lung cancer cells via inhibiting the JAK2/STAT3 signaling pathway. Oncol Rep 2015, 34, 3311-3317, doi:10.3892/or.2015.4279.
  8. Lu, Z.J.; Ren, Y.Q.; Wang, G.P.; Song, Q.; Li, M.; Jiang, S.S.; Ning, T.; Guan, Y.S.; Yang, J.L.; Luo, F. Biological behaviors and proteomics analysis of hybrid cell line EAhy926 and its parent cell line A549. J Exp Clin Cancer Res 2009, 28, 16, doi:10.1186/1756-9966-28-16.

Round 2

Reviewer 1 Report

The Authors have answered to all the request made by the Reviewer.

I have no more academic questions.

Reviewer 2 Report

The authors have adequately addressed the reviewer's concern.